# Analysis of the Relationship between Satisfaction with the Coach and the Effect of Comparative Social Feedback in Elite Female Handball Players

**DOI:** 10.3390/ijerph19137680

**Published:** 2022-06-23

**Authors:** Juan Antonio García-Herrero, Diego Soto-García, Rodrigo J. Carcedo, Isidoro Martínez-Martín, Pedro Delgado-Floody

**Affiliations:** 1Department of Didactics of Musical, Plastic and Corporal Expression, University of Salamanca, 37008 Salamanca, Spain; gherrero@usal.es; 2Department Physical and Sport Education and Group Research AMRED, University of León, 24007 León, Spain; imarm@unielon.es; 3Department of Developmental and Educational Psychology, University of Salamanca, 37005 Salamanca, Spain; rcarcedo@usal.es; 4Department of Physical Education, Sport, and Recreation, Universidad de La Frontera, Temuco 4780000, Chile; pedro.delgado@ufrontera.cl

**Keywords:** motivation, competence, well-being, performance

## Abstract

This research aims at studying the effect of comparative feedback on psychological variables (competence valuation, perceived competence, autonomous motivation, amotivation, subjective well-being) and performance (throwing speed and accuracy). A total of 73 handball players from the highest Spanish handball (Iberdrola League) category participated in this study. After previously rating satisfaction with their head coach, they were indiscriminately assigned to one of three different experimental conditions measuring feedback, positive, negative, and none. There were significant differences in competence valuation, perceived competence, autonomous motivation, and throwing speed in the three feedback groups, more concretely, low satisfaction with the head coach. Positive effects were found when there was low satisfaction with the coach and positive feedback on the competence valuation, autonomous motivation, and throwing speed compared to negative or no feedback. These results have important implications for optimizing coaches’ behaviors in relation to athlete well-being and performance.

## 1. Introduction

Sports training, especially high-performance training, has notably evolved in the search to favor variables that reinforce and optimize athletes’ skills. The coach and athlete need each other to evolve and succeed; this essential process in training fundamentally involves both [1]. Among other abilities, the coach’s feedback to athletes has been related to different effects on performance and psychological variables that affect well-being [2]. Likewise, the coach influences athletes’ behavior and their emotional state, a situation that is key to improving performance [3,4,5]. In this line, one of the key points coaches appreciate is that their athletes are satisfied with their behavior. They consider that the coach fulfills their professional and interpersonal role [6]. Also, coach feedback provided to the athlete can positively affect motor skills and other psychological aspects that determine performance [2]. This work studies the relationship between the perception of satisfaction with the coach and the effect of comparative social feedback on some psychological variables and performance in elite female handball players. 

The athlete’s construction of the perception of the coach’s abilities is formed multidimensionally [7]. Smoll and Smith [8] mention that the final effects that the coach’s behavior exerts are mediated by the significance that the athletes attribute to them. In this sense, Horn [2] points out that the athlete’s evaluation of the coach’s behavior will determine its effectiveness. Myers [9] associated satisfaction with the coach with the perception of competence that the athletes have of them. In other words, the greater the perception of the different abilities, the greater the athlete’s satisfaction with their coach.

On the other hand, the justice perceived by the athlete in the decisions that affect them is an important variable that explains satisfaction with the coach [10,11]. In the same way, Giske, Stein, Johansen, and Høigaard [12] indicated in a study with elite ice hockey and handball players that satisfaction with the coach was predicted by playing time and the perception of justice. It is also convenient to mention that the variables that affect satisfaction can vary based on the different events between them during the season [11,13].

Different studies have shown that high player satisfaction with the coach’s work, and its behavioral benefits, are associated with commitment and performance levels [14,15,16]. In the same sense, Jowett [1] indicates that the relationship between the coach and athlete becomes the means that satisfies, comforts, and supports the improvement of their sports experience, performance, and well-being. It would be interesting to delve into how satisfaction with the coach could influence the effect of feedback on psychological and performance variables in athletes.

Feedback provided to the athlete has received wide attention from the scientific community. Specifically, comparative social feedback establishes performance and personal attribute evaluations in relation to other group members [17]. It is important to point out that the results concerning the effect of feedback on performance are less consistent. Administering different types of feedback has not resulted in differences in performance in some studies [18,19,20]. Specifically, positive feedback in non-expert individuals increases performance [21,22]. In the case of negative feedback, an increase in performance in non-expert individuals has been found; nevertheless, it is important to point out that in this case, the person who provided feedback was not the coach of the high-level athletes who made up the sample, but rather an unrelated researcher [23].

Studies related to the coach’s feedback and behavior have demonstrated that the training conditions that produce positive feelings related to the participants’ results can increase perceived competence and satisfaction [19,24]. Several studies have shown that feedback may influence perceived competence [25,26] and motivation [27]. In non-expert individuals, it has been verified that competence valuation, perceived competence, autonomous motivation, subjective vitality, and throwing speed are favorably influenced by positive feedback [22]. Different studies agree that negative feedback can also have unwanted consequences on motivation, self-esteem, perceived self-efficacy, or the coach-athlete relationship [28,29,30]. In contrast, change-oriented feedback during training is linked to positive effects on different psychological variables that measure the athlete’s well-being [31].

Therefore, this study aimed to analyze the relationship between the satisfaction that elite female players have with their coach and the types of comparative social feedback (positive, negative, and no feedback), and their effects on the perception of competition, motivation (autonomous and amotivational), well-being and performance in a handball throwing task. Thus, based on previous studies that analyzed performance and psychological variables, we expect that satisfaction with the coach modulates the effect of feedback provided to players.

## 2. Materials and Methods

### 2.1. Participants

A total of 73 handball players participated in the study, although only 70 completed the questionnaire and the throwing ball task (Age: *M* = 23.13 years, *SD* = 4.05). Inclusion criteria were a player in the highest category of Spanish handball (Iberdrola League), a field player, and being of legal age. Exclusion criteria were being a goalkeeper and being injured. Each participant had experience in the specific, prescribed task. The participants were only informed that they would participate in a performance task. They were then assigned a number and were randomly assigned to one of three feedback groups: positive (*n* = 26), negative (*n* = 22), and no-feedback (*n* = 22). The coaching staff of the participating club granted permission to conduct the research. The experimental design was conducted according to the ethical standards of the Declaration of Helsinki and approved by the Ethics Committee of the University of León (ETICA-ULE-010-2021).

### 2.2. Apparatus and Task

The selected task (Figure 1) was similar to that used in the study by García-Herrero et al [22]. The participants were asked to throw a ball with as much force and precision as possible with the goal of hitting a cross created by stringing two rubber bands across a handball goal (2.50 m × 1.75 m), nine meters from the throwing area. An official (International Handball Federation) number 2 ball with a mass of 325–400 g and a circumference of 54–56 cm was used for the study.

A radar gun Sports Radar SR3600 (LTD, Homosassa, FL, USA) with ±0.44 m/s was used to record the speed of the ball in each throw [32]. The radar was placed behind the player and pointed in the direction of the target located inside the goal (Figure 1). 

A digital Panasonic SDR-H80 (Panasonic Corp., Osaka, Japan) camera was placed opposite the goal at a distance of 9 m from the goal line and a height of 1.75 m. The center of the ball as it entered the goal was digitalized by the computer software “Kinovea©”, which identified the deviation of the shots with respect to the goal. The point at which the ball entered the goal was indicated digitally. The coordinates of the actual position (for the deviation in the X and Y axis) were calculated using the dimensions of the goal as a reference. Accuracy was measured by the mean radial error (MRE) [33]. The MRE was obtained by digitizing and transforming the throws at the goal into physical coordinates.

### 2.3. Procedure 

On the first day, the players completed the questionnaire. Subsequently, they received information about the task and implemented a standardized warm-up of 12 minutes before the test. To measure the maximum throwing speed, each participant took three jumping maximal throws at the goal without any instruction regarding accuracy and a minute of rest between each shot. The throw with the highest speed was selected from the three attempts. Following this, participants made 21 throws at the goal. The instruction given to the participants was: “throw the ball with as much strength and precision as possible”. The positive and negative feedback groups received feedback every three attempts regardless of their actual performance (the first feedback was received after the third throw). The group without feedback performed 21 throws without receiving any feedback. The type of feedback was positive: “With throws like those, you will be one of the best” or “you’re deviating very little, you’re doing very well” (positive feedback group), and negative: “with shots like those you will be one of the worst?” or “you’re deviating a lot, you’re performing pretty badly” (group of negative comments). The players’ coaches were responsible for providing feedback. The researchers gave them instructions on how to provide feedback. Participants were not allowed to be present when another person took the test. At the end of the task, all participants were informed that the feedback they received was pre-established and did not necessarily coincide with their performance. Before and after completing the task, the participants completed a questionnaire in a private room to evaluate the different psychological variables analyzed in the study.

### 2.4. Measures

#### 2.4.1. Independent Variable

Type of feedback. 

Three types of feedback were included in this study: positive, negative, and no feedback.

Satisfaction with the coach. 

This variable was evaluated with an adaptation of the APCCS II-HST [9] in its Spanish version. This instrument consists of three items (e.g., “how much does your coach know about this sport?”) and represents a reduced version adapted to the Spanish context [10]. The response format is a Likert scale from 1 (very little) to 5 (a lot). Internal consistency reliability was acceptable in this case (α = 0.67).

#### 2.4.2. Psychological Variables


(1)Competence Valuation.


Competence valuation assesses how individuals value good performance on an upcoming task. The variable was measured with a three-item scale (Cronbach’s α = 0.80 before the throwing task and a Cronbach’s α = 0.83 after the throwing task; ICC = 0.73). The two items used by Elliot [34] were also included.
(2)Perceived Competence.

An adaptation of the five items from the corresponding subscale of the Intrinsic Motivation Inventory [35] was used to evaluate participants’ perceptions of competence in relation to the task (Cronbach’s α = 0.78 before the throwing task and Cronbach’s α = 0.83 after the throwing task; ICC = 0.61).
(3)Autonomous Motivation.

An adaptation of the Autonomous Motivation subscale from the Spanish version of the Echelle de Motivation dans les Sports (EMS, Spanish version) [36] was used to measure this construct. This subscale was selected from the study by Mouratidis [19]. The autonomous motivation score (Cronbach’s α = 0.86 before the throwing task and Cronbach’s α = 0.92 after the throwing task; ICC = 0.81) was determined by averaging intrinsic and identified motivation scores.
(4)Amotivation.

An adaptation of the Amotivation subscale from the Spanish version of the Echelle de Motivation dans les Sports (EMS) [36] was used to measure this construct. This subscale was selected from the study by Mouratidis [19]. The amotivation score (Cronbach’s α = 0.79 before the throwing task and Cronbach’s α = 0.80 after the throwing task; ICC = 0.73) was determined by averaging intrinsic and identified amotivation scores.
(5)Subjective Well-Being.

This variable was measured across two dimensions: positive and negative affectivity. Both dimensions were assessed using the Spanish version of the Positive and Negative Affectivity Scale (PANAS; [37]—original version: [38]). This scale consists of 20 items that describe feelings and emotions, of which 10 describe positive affectivity (e.g., enthusiasm); another 10 items measure negative affectivity (e.g., irritable). A total score was obtained by adding the individual scores and dividing them by the number of items answered. Possible scores ranged from 1 (= none) to 5 (= a lot). Good levels of reliability were obtained both in positive affect (Cronbach’s α = 0.87 before the throwing task, and α = 0.89 after the throwing task; ICC = 0.69) and negative affect (Cronbach’s α = 0.82 before the throwing task, and α = 0.86 after the throwing task; ICC = 0.63).

#### 2.4.3. Performance Variables


(1)Throwing Speed.


This variable was recorded in km/h for all 21 throws. The absolute value of the velocity t in km/h of each participant’s shot was divided by their maximum throwing speed and multiplied by 100 to calculate the percentage.

To measure individual maximum throwing speed, each participant performed three jumping maximal throws at the goal without any instructions on accuracy with one minute of rest between each throw. The throw with the highest speed out of the three attempts was chosen. 

Finally, the performance percentage related to maximum throwing speed was used to measure this variable. The 21 throws were divided into three sets: 1–3 (no feedback for any group), 4–12 (positive and negative feedback groups started receiving feedback), and 13–21 (comparative feedback was increasing for both groups). A total score for each set was obtained by adding the individual performance percentages related to the maximum throwing speed of each shot and dividing them by the number of throws.
(2)Throwing Accuracy.

The MRE was used to measure throwing accuracy. The MRE was determined by measuring the average absolute distance from the center of the target of the 21 throws. Again, the 21 throws were divided into three sets (1–3, 4–12, and 13–21). A total score for each set was obtained by adding the individual throwing accuracy scores of each shot and dividing them by the number of throws.

### 2.5. Data Analysis

An α-level of 0.05 was employed for all analyses. Violations of normality and variance homogeneity in all repeated measures ANOVA models, the small sample size, and the use of ordinal Likert-type scales data, required a nonparametric approach [39,40,41] using an f2-ld-f1 function in the software package “nparLD” [42] included in “R 4.1.2” (R. Estudio, Boston, EE.UU). 

In the case of significant interaction effects, post hoc pairwise comparisons were contrast effects of psychological variables. Performance between the three types of feedback for each time of measure was calculated using the function nparcomp of the R package “nparcomp” [43,44]. Post hoc pairwise comparisons between the different times of measure for each type of feedback group was tested using a nonparametric studentized permutation analysis with 10,000 repetitions (function npar.t.test.paired of the R package “nparcomp”) and a Holm–Bonferroni correction for multiple comparisons [45]. Finally, Cliff’s Delta was used to measure the nonparametric effect size of pairwise comparisons using the R package “effsize” [46].

## 3. Results

Seven nonparametric ANOVA with one sub-plot factor containing two levels for psychological variables (before and after the throwing task) and three levels for performance variables (1–3 throws, 4–12 throws, and 13–21 throws), and two whole-plot factors with three levels for feedback (positive, negative, and lack of feedback) and two levels for satisfaction with the coach (low and high) were performed to study the impact of these factors on psychological variables such as competence (competence valuation and perceived competence), motivation (autonomous motivation and amotivation), subjective well-being (positive and negative affect) (see Table 1), and performance variables such as throwing speed and throwing accuracy (see Table 2).

### 3.1. Psychological Variables

#### 3.1.1. Competence

Feedback main effect and satisfaction x feedback interaction were significantly associated with competence value. Overall, the positive feedback group showed higher levels of competence than the negative feedback group (Δ = 0.27, *p* < 0.05). Regarding satisfaction × feedback interaction, participants in the low satisfaction with the coach and positive feedback groups showed higher competence value levels than those with low satisfaction with the coach but were included in the negative (Δ = 0.72, *p* < 0.001) or no feedback groups (Δ = 0.66, *p* < 0.01). However, no differences between the feedback groups were found in the high satisfaction with the coach group.

Additionally, a significant interaction of time x feedback was found for perceived competence. Those who received positive or negative feedback showed higher and lower competence levels after the throwing task, respectively, compared to the evaluation before the task (Δ = 0.26, *p* < 0,05; Δ = 0.28, *p* < 0.01). No changes were observed in the no-feedback group. 

#### 3.1.2. Motivation

The second-order interaction satisfaction with the coach x feedback x time was significant. Post-hoc tests revealed that those with low satisfaction with the coach who received positive feedback showed higher levels of autonomous motivation after the throwing task than before (Δ = 0.22, *p* < 0.05). No other time differences were found in the other groups formed as a combination of the levels of satisfaction with the coach and feedback (see Table 1).

Regarding amotivation, the main effects of satisfaction with the coach and feedback were significant. Participants in the low satisfaction with the coach group showed more amotivation than the group with high satisfaction (Δ = 0.21, *p* < 0.05). Also, those in the no-feedback group showed higher levels of amotivation than those in the negative feedback group (Δ = 0.35, *p* < 0.01), and lower levels of amotivation than those in the positive feedback group (Δ = 0.41, *p* < 0.001).

#### 3.1.3. Well-Being

No significant results were found in the case of positive and negative affect outcomes.

#### 3.1.4. Performance Variables

The interaction satisfaction with the coach x time (1–3 throws, 4–12 throws, and 13–21 throws) was significant. Post hoc tests indicated that participants in the low satisfaction with the coach group decreased their throwing speed from the 1-3 throws set to the 4–12 throws set (Δ = 0.24, *p* < 0.05) and to 13-21 throws (Δ = 0.30, *p* < 0.05). No differences were found in the groups with high satisfaction with the coach.

No significant results were found for throwing precision.

**Table 1 ijerph-19-07680-t001:** Descriptives and nonparametric repeated measures ANOVA-type models of psychological variables for each type of feedback, satisfaction with the coach, time of measure, and their interactions.

	Low Satisfaction									High Satisfaction									ANOVA-Type			Significant Interactions Post Hoc Comparisons ^b^
	F0			F+			F−			F0			F+			F−						
	Mdn	M	n	Mdn	M	n	Mdn	M	n	Mdn	M	n	Mdn	M	n	Mdn	M	n	Effect	F	df ^a^	
Value Competence																						
Time 1	5.00	4.70	7	6.75	6.39	11	5.00	5.16	6	5.75	5.73	15	5.75	5.65	15	5.62	5.64	16	S	2.17	1.00	
Time 2	4.75	4.75	7	6.75	6.18	11	4.50	4.50	6	5.75	5.82	15	5.75	5.73	15	6.00	5.92	16	F	4.40 **	1.90	
																			T	0.08	1.00	
																			S×F	5.65 **	1.90	LS: F+>F−; F+>F0
																			S×T	3.06	1.00	
																			F×T	0.01	1.72	
																			S×F×T	0.28	1.72	
Competence																						
Time 1	4.80	4.77	7	4.80	4.42	11	4.40	4.50	6	3.60	3.96	15	4.60	4.32	15	4.90	4.53	16	S	0.82	1.00	
Time 2	4.40	4.20	7	4.60	4.56	11	4.30	4.26	6	3.60	3.80	15	5.00	4.90	15	4.40	3.75	16	F	1.36	1.97	
																			T	1.50	1.00	
																			S×F	0.94	1.97	
																			S×T	0.61	1.00	
																			F×T	4.08 *	1.88	F+: t_1_<t_2_ F−: t_1_>t_2_
																			S×F×T	1.22	1.88	
Autonomous Motivation																						
Time 1	5.75	5.47	7	5.59	5.61	11	5.71	5.74	6	5.92	5.72	15	5.58	5.41	15	6.00	5.96	16	S	0.38	1.00	
Time 2	5.33	5.20	7	5.75	5.86	11	5.84	5.80	6	5.92	5.79	15	5.58	5.38	15	6.04	5.99	16	F	1.09	1.98	
																			T	0.64	1.00	
																			S×F	1.27	1.98	
																			S×T	0.01	1.00	
																			F×T	1.02	1.78	
																			S×F×T	3.33 *	1.78	LS and F+: t_1_<t_2_
Amotivation																						
Time 1	3.59	3.68	7	2.00	2.52	11	2.38	2.63	6	2.50	2.62	15	2.25	2.52	15	1.88	2.44	16	S	5.54 *	1.00	
Time 2	5.33	4.03	7	1.75	2.16	11	2.13	2.54	6	2.50	2.48	15	2.00	1.97	15	1.75	2.20	16	F	7.46 ***	1.86	
																			T	2.78	1.00	
																			S×F	1.84	1.86	
																			S×T	0.94	1.00	
																			F×T	1.85	1.86	
																			S×F×T	0.29	1.86	
Positive Affect																						
Time 1	3.70	3.67	7	3.90	4.06	11	3.75	3.77	6	3.70	3.65	15	3.60	3.54	15	3.90	3.83	16	S	0.06	1.00	
Time 2	3.30	3.41	7	4.10	4.09	11	3.60	3.57	6	3.70	3.69	15	3.70	3.81	15	3.80	3.71	16	F	1.17	1.96	
																			T	0.51	1.00	
																			S×F	1.17	1.96	
																			S×T	1.26	1.00	
																			F×T	1.45	1.64	
																			S×F×T	0.32	1.64	
Negative Affect																						
Time 1	1.30	1.59	7	1.30	1.33	11	1.25	1.22	6	1.20	1.41	15	1.20	1.37	15	1.45	1.69	16	S	0.41	1.00	
Time 2	1.50	1.75	7	1.30	1.30	11	1.20	1.21	6	1.40	1.44	15	1.20	1.35	15	1.45	1.81	16	F	0.69	1.93	
																			T	0.52	1.00	
																			S×F	1.81	1.93	
																			S×T	0.41	1.00	
																			F×T	0.04	1.95	
																			S×F×T	0.17	1.95	

Note: S = Satisfaction; LS = Low Satisfaction with Coach; HS = High Satisfaction with Coach; F = Feedback; F0 = No Feedback; F− = Negative Feedback; F+ = Positive Feedback; T = Time; t_1_ = Time 1 (before the task); t_2_ = Time 2 (after the task); Q = Quartile. ^a^ The denominator of all df values is ∞; e.g., 1.96, ∞. ^b^ Significant post-hoc pairwise comparisons using Holm-Bonferroni adjustment. α-level is set at 0.05; * *p* < 0.05, ** *p* < 0.01, *** *p* < 0.001.

**Table 2 ijerph-19-07680-t002:** Descriptives and nonparametric repeated measures ANOVA-type models of performance variables for each type of feedback, satisfaction with the coach, time of measure, and their interactions.

	Low Satisfaction									High Satisfaction									ANOVA-Type			Significant Interactions Post Hoc Comparisons ^b^
	F0			F+			F−			F0			F+			F−						
	Mdn	M	n	Mdn	M	n	Mdn	M	n	Mdn	M	n	Mdn	M	n	Mdn	M	n	Effect	F	df ^a^	
Throwing Speed																						
Time 1	87.92	87.87	7	87.67	87.67	11	89.86	87.12	6	84.09	85.04	15	83.57	85.10	15	87.70	86.38	16	S	0.13	1.00	
(1–3)																			F	0.55	1.97	
Time 2	83.68	85.82	7	86.83	87.20	11	84.05	82.34	6	84.86	84.86	15	85.90	86.91	15	86.18	86.97	16	T	4.70 *	1.55	
(4–12)																			S×F	0.97	1.97	
Time 3	83.54	85.52	7	86.55	87.27	11	86.55	87.27	6	84.81	84.04	15	85.51	86.60	15	85.00	86.66	16	S×T	5.10 *	1.55	LS: F+>F−; F+>F0
(13–21)																			F×T	2.22	2.79	
																			S×F×T	0.26	2.79	
Throwing Accuracy																						
Time 1	67.59	85.92	7	40.92	63.72	11	50.39	66.58	6	60.64	59.82	15	45.00	62.71	15	44.17	62.49	16	S	0.18	1.00	
(1–3)																			F	1.58	1.85	
Time 2	64.21	77.52	7	41.57	66.27	11	51.18	61.73	6	46.68	50.15	15	54.65	57.71	15	47.08	59.24	16	T	1.08	1.76	
(4–12)																			S×F	1.58	1.85	
Time 3	61.52	81.24	7	43.78	59.16	11	43.78	59.16	6	48.50	55.49	15	42.36	53.85	15	51.17	61.37	16	S×T	0.66	1.76	
(13–21)																			F×T	1.04	3.17	
																			S×F×T	1.65	3.17	

Note: S = Satisfaction; LS = Low Satisfaction with Coach; HS = High Satisfaction with Coach; F = Feedback; F0 = No Feedback; F− = Negative Feedback; F+ = Positive Feedback; t = Time; t_1_ = Time 1 (1–3 pitches); t_2_ = Time 2 (4–12 pitches); t_3_ = Time 3 (13–21 pitches); Q = Quartile. ^a^ The denominator of all df values is ∞; e.g., 1.96, ∞. ^b^ Significant post-hoc pairwise comparisons using Holm-Bonferroni adjustment. α-level is set at 0.05; * *p* < 0.05.

## 4. Discussion

The aim of this study was to analyze the effect of comparative social feedback on psychological and performance variables related to satisfaction with their coach. According to previous studies [22,46], positive feedback can increase the athletes’ perception of competence. Although, in general, these studies were developed with inexperienced participants where it was frequent that the person offering the information about the task was not the athlete’s coach. The results in this work confirm that in elite players, positive feedback provided by the coach increased their perception of competence. In contrast, negative feedback (also provided by their coach) generated a significant decrease in the players’ perception of competence. These results differ from those of other studies in which the researcher offered feedback to high-level players [23]. Unlike our results, this positive, negative, or no feedback information provided by someone not associated with the players did not significantly affect their perception of competence. The information provided to high-level athletes may not have the same effect, depending on who provides it.

Regarding the effect of feedback, our results show differences when the players had a high or low level of satisfaction with the coach. In the latter, where satisfaction with the coach was low, and players received positive feedback, their autonomous motivation significantly increased at the end of the intervention compared to the beginning. These results have not been seen in female players with high satisfaction with their coach, in who the effect of feedback (positive, negative, or no feedback) did not generate any change in their autonomous motivation. Likewise, the results of female players with low satisfaction with their coach showed that positive feedback significantly increased the value given to the task compared to negative feedback or the group with no feedback. As before, female players with high satisfaction with their coach did not show any change in this variable. Based on these results, it seems that the effect of positive feedback was greater on autonomous motivation and the value they gave to the task when the female players had a low perception of their coaches’ competence. It can be seen how the information provided by the coaches had different effects depending on the satisfaction that the female players had with them. In this sense, it has been identified that when satisfaction with the coach is high, the type of feedback has not generated any change in the psychological variables studied (value competence, competence, autonomous motivation, amotivation, positive affect, negative affect).

In contrast, when satisfaction with the coach was low, positive feedback changed autonomous motivation and value given to the task. In the case of athletes in training who assessed their coach’s level of competence, a variable associated with satisfaction with the coach [9], none of the three types of feedback provided affected autonomous motivation [47]. Our results are consistent with those of Amorose and Nolan [48], in which the importance that the players attributed to their coach conditioned the results of different variables. It is possible that the effect of feedback provided to the athletes is modulated by the degree of satisfaction that the athletes have with their coaches. 

For the performance variables, speed and precision, significant differences were found in the players in those who received positive feedback and had low coach satisfaction in time 3 (13–21 throws). In other words, players who were dissatisfied with their coach and received positive feedback were able to shoot faster than those who received negative or no feedback. These results coincide with those in inexperienced subjects who received positive feedback and improved their performance [18,20] and with another study where individuals received different types of comparative feedback, and only those with positive feedback improved compared to negative and none [22]. In contrast, with expert female players, the effect of negative feedback provided by the researcher showed an improvement in throwing speed [23]. Again, it is surprising to find that high satisfaction with the coach does not affect performance variables regardless of the type of feedback provided by the coach. 

Finally, the results reached in this research could be due to the credibility and relationship that the elite female players have with their coach; therefore, the effect of the different types of feedback does not generate direct effects but rather is conditioned by their perception of the coach.

### Limitations and Future Research

As in all studies, this one has limitations. The effects of feedback and satisfaction with the coach were studied only in a specific task. Future studies should study the effect of applying comparative feedback during longer periods, for example, during an entire season, and analyze how satisfaction with the coach can influence psychological and performance variables. Likewise, future studies of the effect of feedback should also be performed during real competition with adversaries using methodologies different from those in this study, increasing the ecological validity. Finally, the participants’ gender could have influenced these results; therefore, it would be necessary to study the effects of feedback on the behavior of elite male players. 

## 5. Conclusions

In light of the results in this study of elite female handball players, low satisfaction with the coach and comparative positive feedback generate significant changes in the value given to the task, autonomous motivation, and throwing speed. In contrast, positive feedback did not affect psychological and performance variables in athletes who showed high satisfaction with their coach. Also, without considering the level of satisfaction with the coach, positive feedback generates a player’s greater perception of competence than negative feedback.

In conclusion, in female elite handball players that show low satisfaction with their coach, the feedback provided affects the psychological and performance variables, unlike those who are highly satisfied, who do not show any significant variation.

Based on these findings, we recommend that coaches provide positive comments to their players, especially those who may be unsatisfied with their coach’s work due to the different events that occur during the season.

## Figures and Tables

**Figure 1 ijerph-19-07680-f001:**
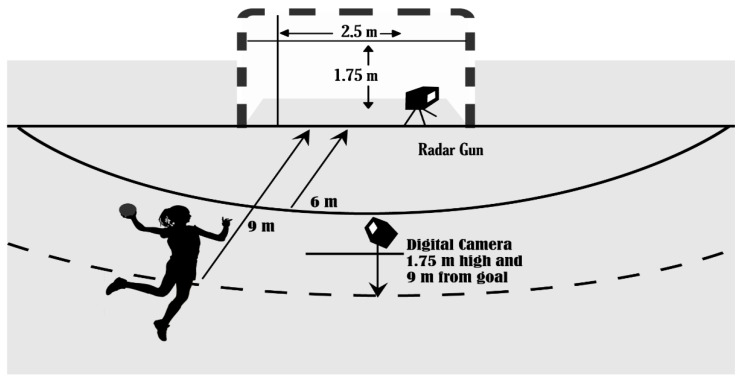
Graphical representation of the ball throwing task.

## Data Availability

The data presented in this study are available on request from the corresponding author. Data are not publicly available because they will be used in combination with other current researchs.

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
