# Peer review of "Analysis of the Relationship between Satisfaction with the Coach and the Effect of Comparative Social Feedback in Elite Female Handball Players"

_ijerph, 2022, doi:10.3390/ijerph19137680_

Round 1

Reviewer 1 Report

Dear Authors,

It was interesting reading and I did it whit pleasure. Please look at the pdf file and comments related to your paper.

The main concerns are about a few parts similar to the previous studies, and two literature positions without detailed citation information.

Author Response

Dear Reviewer

First of all, thank you for your comments.

Please note that we have responded to your comments within the pdf file and you will be able to see the modifications in the new version.

Thank you very much for your comments regarding the results.  We have modified your presentation and the mistakes.

In reference to your comment "The main concerns are about a few parts similar to the previous studies" we agree with it and part of the methodology is similar to previous researchs but let us point out the following:

1.- In the others reasearchs the feedback is done by a researcher, so there are no previous links.

In this reresearch the coaches were in charge of providing feedback to the players.

2º.- The throwing task was designed to be more applied to what happens in competition. Jumping distance throwing has a lot of transfer to the real game.

3º.- The sample is exclusively elite women, another element that differentiates both works.

We hope to have been able to clarify the differences between the two studies and the need to continue deepening the knowledge of how feedback affects the players.

Reviewer 2 Report

The research presented by the authors shows a high interest in the ecological characteristics of the research. They have participated in the study of coaches and elite athletes which provides an added value.

The research design is appropriate for the proposed objectives.

It shows the most relevant theories.

The statistical analysis is ideal.

The discussion delves into the effects of comparative feedback on elite female handball players.

General considerations:

In your paper, you indicate that the feedback is provided by the coach, but it is interesting to know if it was your head coach or an assistant.

 Line 59-65. In the paper, you indicate satisfaction is related to performance and psychological benefits. Could you explain it better?

 Line 79-85. Indicate the results with non-expert and expert players. Could you be more specific and indicate if they are male or female?

 Line 103. Clarify if they had had experience in the task or if they had experience as an athlete. It is not the same thing.

 Line 145-149. What was the procedure for female players who do not receive feedback?

 Line 340. It would be important to indicate whether the results achieved are men or women.

Author Response

Dear reviewer

First of all, thank you for your comments.

Please note that within the text you will find the corrections to your comments.

Additionally, additional information is included in the results and they are presented in a clearer way.

You can see all the changes highlighted in red color within the document.
